# Observation of bosonic condensation in a hybrid monolayer MoSe$_2$-GaAs microcavity

Max Waldherr[1], Nils Lundt[1], Martin Klaas[1], Simon Betzold[1], Matthias Wurdack[1], Vasilij Baumann[1], Eliezer Estrecho[2], Anton Nalitov[3,4,5], Evgenia Cherotchenko[4,5], Hui Cai[6], Elena A. Ostrovskaya[2], Alexey V. Kavokin[5,7,8], Sefaattin Tongay[6], Sebastian Klembt[1], Sven Höfling[1,9] & Christian Schneider[1]

Bosonic condensation belongs to the most intriguing phenomena in physics, and was mostly reserved for experiments with ultra-cold quantum gases. More recently, it became accessible in exciton-based solid-state systems at elevated temperatures. Here, we demonstrate bosonic condensation driven by excitons hosted in an atomically thin layer of MoSe$_2$, strongly coupled to light in a solid-state resonator. The structure is operated in the regime of collective strong coupling between a Tamm-plasmon resonance, GaAs quantum well excitons, and two-dimensional excitons confined in the monolayer crystal. Polariton condensation in a monolayer crystal manifests by a superlinear increase of emission intensity from the hybrid polariton mode, its density-dependent blueshift, and a dramatic collapse of the emission linewidth, a hallmark of temporal coherence. Importantly, we observe a significant spin-polarization in the injected polariton condensate, a fingerprint for spin-valley locking in monolayer excitons. Our results pave the way towards highly nonlinear, coherent valleytronic devices and light sources.

[1] Technische Physik and Wilhelm-Conrad-Röntgen-Research Center for Complex Material Systems, Universität Würzburg, Am Hubland, 97074 Würzburg, Germany. [2] ARC Centre of Excellence in Future Low-Energy Electronics Technologies and Nonlinear Physics Centre, Research School of Physics and Engineering, The Australian National University, Canberra, ACT 2601, Australia. [3] Science Institute, University of Iceland, Dunhagi 3, 107 Reykjavik, Iceland. [4] ITMO University, St. Petersburg 197101, Russia. [5] Physics and Astronomy School, University of Southampton, Highfield Campus, Southampton SO171BJ, UK. [6] School for Engineering of Matter, Transport, and Energy, Arizona State University, Tempe, AZ 85287, USA. [7] SPIN-CNR, Viale del Politecnico 1, 00133 Rome, Italy. [8] Spin Optics Laboratory, St-Petersburg State University, 1, Ulianovskaya 194021, Russia. [9] SUPA, School of Physics and Astronomy, University of St. Andrews, St. Andrews KY16 9SS, UK. These authors contributed equally: Max Waldherr, Nils Lundt, Martin Klaas. Correspondence and requests for materials should be addressed to C.S. (email: Christian.Schneider@physik.uni-wuerzburg.de)

Bosonic condensation is an intriguing phenomenon, which describes the collective collapse of quantum particles into a single macroscopic and coherent quantum state. For a long time, experiments devoted to bosonic condensation have been reserved for ultra-cold atoms[1,2], but more recently became accessible in open-dissipative solid-state systems[3] at elevated temperatures. A prime candidate to observe bosonic condensation in solids are exciton-polaritons, which are bosonic quasi-particles resulting from strong light–matter coupling in micro-cavities with embedded materials that are characterized by a large dipole oscillator strength[4,5]. These composite particles possess a variety of very appealing physical properties, particularly prominent at larger densities. They are bosons with a very low and tunable effective mass and are therefore almost ideally suited for studies of Bose–Einstein condensation phenomena[3,6] at elevated temperatures[7,8].

The intrinsic properties of exciton-polaritons critically depend on properties of the matter excitations. Atomically thin monolayers of transition metal dichalcogenides (TMDCs) have emerged as a new material platform with highly interesting excitonic properties: the materials are mono-atomically thin, thus composing the ultimate physical limit for a system to host collective electronic excitations. They are highly nonlinear, and the chiral exciton properties are uniquely linked to the valley degree of freedom, that is, excitons emerging from the direct band transition at the K-point (K′-point) possess a pseudospin projection to the structure axis of $+1$ ($-1$). Locking of spin and valley index directly protects valley excitons from fast spin relaxation for small exciton momenta[9–11], and, more importantly, it allows to interlink excitonic propagation with its chirality, driven by the Berry curvature of the valleys[12]. This new effect paved the way to the new research area of valleytronics[12–14].

Valleytronics with excitons intrinsically suffers from the very short diffusion length of the optically addressable quasi-particles, which limits the zoo of observable phenomena, or makes them at least very hard to detect with conventional experimental methods. This includes the possible manifestation of chiral excitonic currents on the edge of carefully prepared twisted bilayers[15], the interplay between long-range order[3] and valley physics[12], and the interplay between superconductivity and superfluidity[16,17], which has been predicted in such systems. Cavity exciton-polaritons can, in principle, provide a feasible solution to this roadblock, as the expansion of a polariton cloud in a microcavity and the build-up of its coherence is known to be very fast. In this spirit, it has been shown that the valley-selective strong coupling regime with excitons in atomic monolayers of $MoS_2$[18], $MoSe_2$[19,20], $WS_2$[21], and a microcavity resonance in the linear (low-density) regime is accessible.

However, pronounced polariton expansion over macroscopic distances[22], the formation of robust spin patterns[23], and the emergence of topological excitations[24] are expected in the nonlinear (high-density) regime of bosonic condensation. This regime is anticipated in microcavities with embedded TMDC monolayers, but has not been accessible so far due to experimental limitations, such as the fast recombination rate of excitons in TMDCs and the limited photon lifetime in current microcavities.

Here, we utilize the simultaneous coupling[25] of excitons in GaAs quantum wells (QWs) and $MoSe_2$ to a joint photonic resonance, combining efficient polariton energy relaxation with robust preservation of an optically induced spin-polarization. The hybridization leads to a longer radiative lifetime of exciton-polaritons and their reservoirs[26], and, particularly, it facilitates the build-up of the critical population for bosonic stimulation in the ground state. We detect the condensation of exciton-polaritons into the lowest, quantized energy state of the hybrid mode by

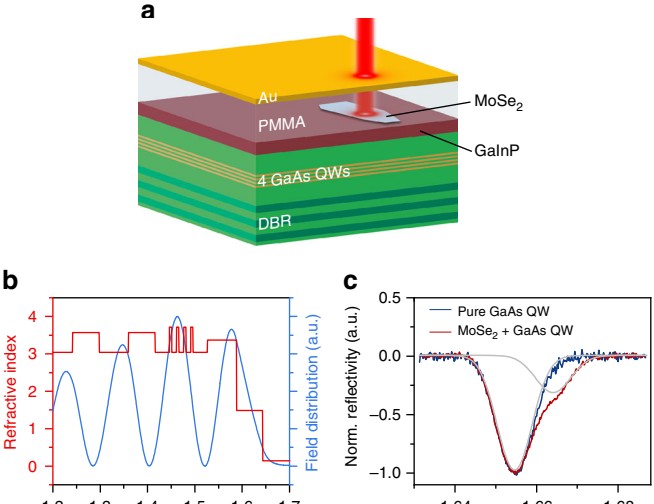

**Fig. 1** Hybrid Tamm monolayer device. **a** Schematic illustration of the Tamm-plasmon device with the embedded $MoSe_2$ monolayer. The monolayer is capped with PMMA, whose thickness primarily determines the frequency of the device's optical resonance. **b** Calculation of the electromagnetic field intensity in the heterostructure. The field distribution in the device is designed to yield optimal overlap with the position of the quantum wells as well as the atomic monolayer. **c** Reflectivity spectrum of the device prior to capping the structure with Au. The two absorption dips are correlated with the GaAs exciton and the $MoSe_2$ neutral excitonic transition

measuring its population and infer the coherence by tracing the spectral width of the emitted light. In the regime of hybrid polariton condensation, we observe clear indications of interactions between polaritons and excitons in the optically injected reservoir by studying the power-dependent energy shift of the mode and confirm that the valley index of the monolayer remains addressable.

## Results

**Technology**. A sketch of the studied photonic microstructure, giving rise to the so-called Tamm-plasmon polariton resonances[27,28], is depicted in Fig. 1a. The structure is similar to the one described in ref.[29]: it consists of an AlAs/AlGaAs distributed Bragg reflector (DBR; 30 pairs), which is characterized spectrally by its stopband ranging from 710 to 790 nm (see ref.[29]), with reflectivity up to 99.9% between 740 nm (1.675 eV) and 765 nm (1.621 eV) at 10 K. The AlAs/AlGaAs Bragg stack, which has been grown by gas source molecular beam epitaxy, is topped with a 112-nm-thick AlAs layer with four embedded GaAs QWs. A layer of GaInP caps the AlAs layer. A single monolayer of $MoSe_2$, mechanically exfoliated via commercial adhesive tape from a bulk crystal, was transferred onto the top GaInP layer with a polymer stamp[30].

The full cavity device is completed by capping the monolayer with an 80-nm-thick layer of polymethyl methacrylate (PMMA) and a 60-nm-thick gold layer, such that it promotes an optical resonance with a field antinode both at the position of the monolayer and at the stack of GaAs QWs (Fig. 1b), and yields an estimated Q-factor of 650. We characterized the absorption of our embedded material in a high-resolution measurement prior to completion of our microcavity at a sample temperature of 10 K (Fig. 1c). There, we traced out two absorption resonances, which we attribute to the free exciton in the $MoSe_2$ monolayer, as well as the GaAs QWs.

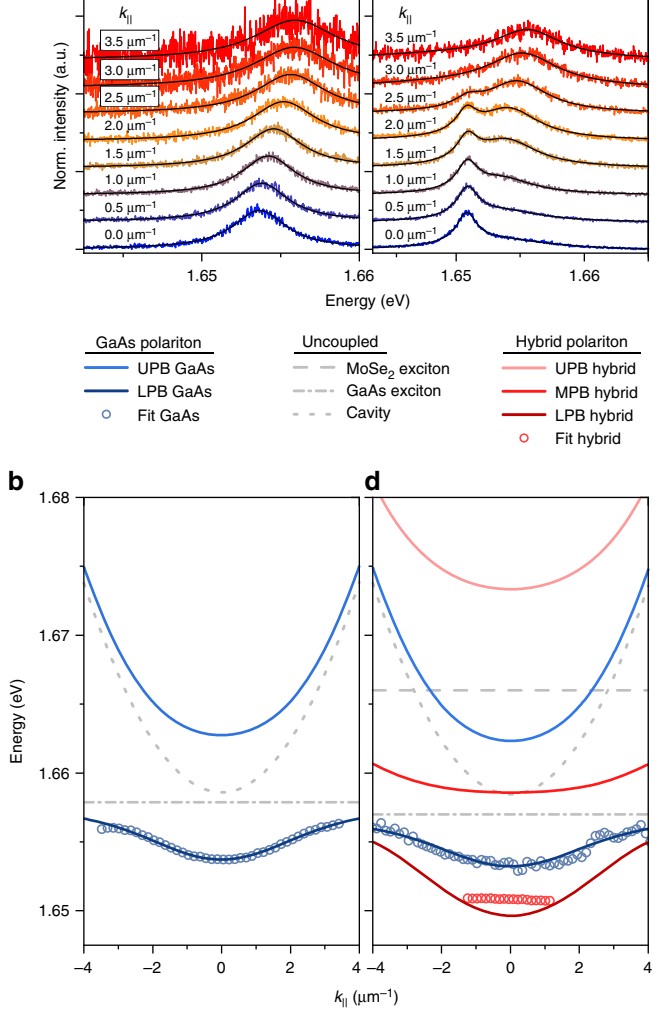

GaAs polariton
— UPB GaAs
— LPB GaAs
○ Fit GaAs

Uncoupled
-- MoSe₂ exciton
-·- GaAs exciton
··· Cavity

Hybrid polariton
— UPB hybrid
— MPB hybrid
— LPB hybrid
○ Fit hybrid

**Fig. 2** Luminescence of GaAs exciton-polaritons and hybrid polaritons. **a** Momentum-resolved photoluminescence spectra recorded from the device at 4.2 K at the periphery of the monolayer, depicted in a waterfall representation. **b** Energy–momentum dispersion relation of the signal, following the model of the lower polariton branch in a coupled oscillator system. **c** Plot of the photoluminescence from the device at the MoSe₂ monolayer position. Two peaks evolve, which are attributed to the hybrid polariton mode and the GaAs polariton resonance from the surroundings of the monolayer. **d** Energy–momentum dispersion relation of the two modes corresponding with the signals shown in **c**. The discrete, hybrid polariton mode is a result of an admixture of 21.0% GaAs, 21.5% MoSe₂, and 57.6% photon (see Supplementary Note 1)

**Hybrid MoSe₂-GaAs exciton–polariton mode**. In order to confirm the emergence of a hybrid polariton mode in our device, we first study the dispersion relation of the bare GaAs QW exciton-polaritons by recording the emission at a position near the monolayer, which is depicted in Fig. 2a. The spectra in the waterfall representation in Fig. 2a were extracted from angle-resolved emission spectra, which we recorded in a modified micro-photoluminescence setup in the far-field imaging configuration (see Methods). We can clearly construct the dispersion relation of the QW exciton-polaritons with a mass of $4 \times 10^{-5}$ times the free electron mass, a Rabi splitting of 9.2 meV, and a positive exciton–photon detuning of 5.9 meV. These observations are fully consistent with previous results reported in a similar device[29].

Next, we record the spectral emission signatures of our sample at the position of the monolayer. Due to the small size of the flake (approx. 1.7 μm in diameter, see below), along with the emission from the monolayer, we also collect signal from its periphery due to the lateral diffusion of GaAs QW excitons. The corresponding spectra shown in Fig. 2c are composed of a high-energy dispersive signal, which follows precisely the dispersion relation of the bare III–V GaAs exciton-polaritons in Fig. 2a and a second, red-shifted signal of a dispersion-less resonance spread by 2.5 μm⁻¹ in the momentum space. That signal is consistent with the formation of a spatially confined hybrid exciton–polariton mode in the collective strong coupling regime between excitons in the MoSe₂ monolayer and GaAs QW, respectively, and the cavity mode. The finite size of the monolayer induces a strong quantization of the hybrid mode yielding a confinement-induced blueshift of 1.21 meV with respect to the energy minimum of the hybrid mode at the lowest excitation power. This dispersion relation was calculated assuming a coupling strength of 20 meV between the monolayer exciton and the Tamm resonance, that is based on our previous findings[29]. By treating the monolayer as a finite potential well with the confinement depth of 2.38 meV, which is given by the difference between the ground state of the GaAs polariton dispersion outside the monolayer and the minimum of the hybrid polariton dispersion, we deduce the lateral extent of the monolayer of approximately 1.7 μm.

**Condensation of polaritons**. The formation of a condensate of exciton-polaritons emerging in our hybrid mode can be visualized in power-dependent experiments, which are carried out at a sample temperature of 4.2 K. In Fig. 3a–d, we plot far-field spectra recorded from the position of the monolayer at various pump powers. Exciton-polaritons are injected in the system by 2-ps-long laser pulses (82 MHz repetition rate). The wavelength is tuned approximately to the energy of the upper hybrid polariton branch at 741 nm. Here, we observe that with growing pump powers the spectrum is progressively dominated by the identified hybrid mode. At the same time, the hybrid mode undergoes a distinct energy blueshift and its spectral width narrows. In Fig. 3e, we provide a more detailed analysis of this primary emission feature. From the plotted input–output curve, we can deduce a clear threshold behavior at pump energies as low as 4.8 pJ per pulse (gray shaded area), which is accompanied by a rapid drop in the polariton linewidth from 2.1 meV down to 0.7 meV. This drop in the linewidth is a strong evidence of the onset of temporal coherence in systems emerging at the transition from a thermal to a coherent state[31]. One crucial difference between polariton condensates and classical laser modes is manifested by the excitonic component of cavity polaritons, which governs the polariton interactions with uncoupled excitons and other polaritons. These interactions induce the blueshift of the polariton mode, which is plotted in Fig. 3f as a function of the energy per injection pulse. The inset depicts the energy shift as a function of the polariton occupancy in the hybrid mode, which is calculated by normalizing the emission intensity by the intensity at the threshold. Below the condensation threshold, the hybrid mode experiences an approximately linear blueshift with increasing pump power on the order of 3 meV, which we attribute primarily to the interaction between the hybrid polaritons and excitons in the non-resonantly driven reservoirs which are continuously built up. As we cross the threshold, the mode continues to blueshift by approximately 300 μeV (also see inset), yet the slope of the curve changes which reflects the modified growth rates of the two reservoirs due to formation of the polariton condensate (see the Supplementary Note 4). This effect can be accounted for by using

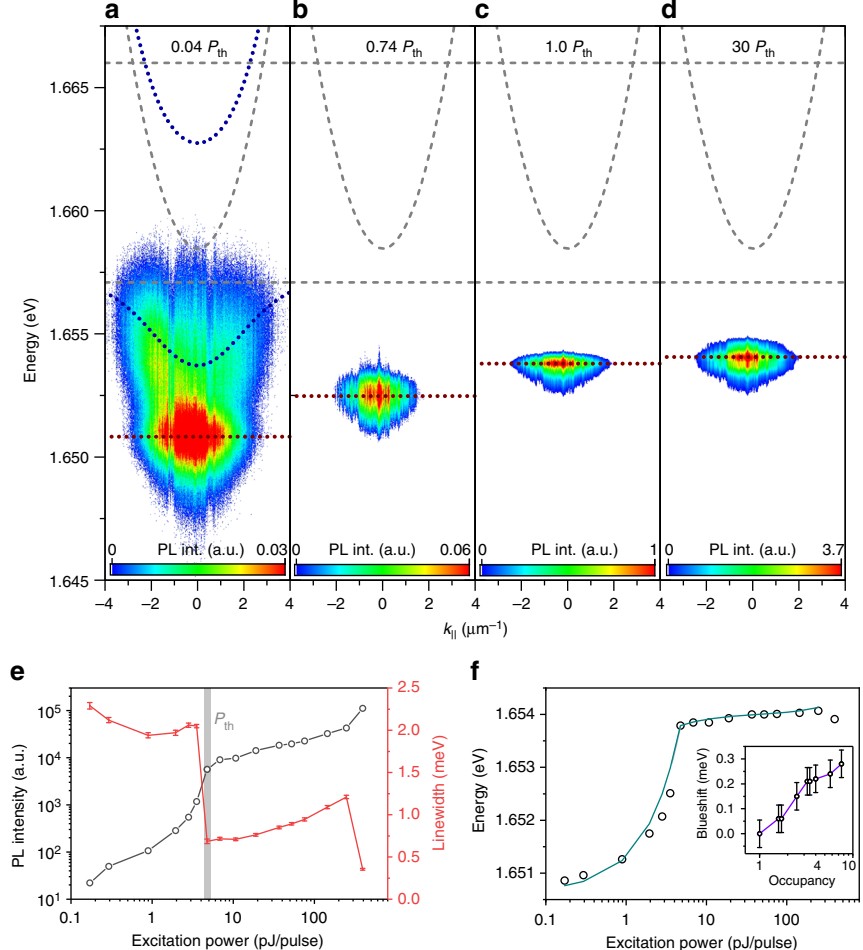

**Fig. 3** Density-dependent characterization of the hybrid exciton-polaritons. **a**–**d** False color intensity profile of the hybrid polariton device at different excitation powers, below (**a**), close to (**b**), at (**c**), and above (**d**) the threshold. The red dotted lines resemble the energy of the mode and serve to illustrate the blueshift at increasing excitation powers. **e** PL emission intensity (black) and linewidth (red) as a function of the excitation power. **f** Blueshift of the hybrid polariton mode across the condensation threshold. Inset: Blueshift above the threshold as a function of the polariton occupancy, which was normalized to unity at the threshold

an analytical model which constitutes a set of semi-classical Boltzmann rate equations accounting for the two excitonic reservoirs (see the Supplementary Note 4) and produces a good fit to our data as shown in Fig. 3f.

We note that, at the largest pump powers, we observe a second deviation from the linear increase of intensity above the condensation threshold (accompanied by a drop in the emission linewidth). This deviation is in general agreement with the strong to weak coupling laser transition for the GaAs polaritons, which was observed in a comparative measurement in the vicinity of the MoSe$_2$ monolayer (see Supplementary Note 2).

**Valley polarization.** Finally, we address the question of whether the valley index of monolayer excitons can be controlled and preserved in a condensate of hybrid exciton-polaritons. Therefore, we drive the system by a circularly polarized injection laser, to inject excitons predominantly in one valley of the embedded monolayer. As our laser is injecting quasi-particles into the upper hybrid mode, we expect that we predominantly create polaritons tagged by one valley index. This is reflected by the circular polarization of the condensate shown in Fig. 4a, b, which clearly retains the polarization of the pump laser to a significant degree. The degree of circular polarization (DOCP) is calculated via $(N^{\pm} - N^{\mp})/(N^{\pm} + N^{\mp})$ and yields a value as high as 17.9% for σ$^+$pumping (16.4% for σ$^-$ pumping) for an average polariton

population at the pumping power of 10 $P_{th}$. The degree of polarization emitted from the hybrid mode is strongly reduced to approximately 9% in the linear regime (Fig. 4a, b). As a reference, the bare GaAs exciton-polaritons show a reduced circular polarization in the linear (low-density) regime (recorded at the comparable laser-exciton detuning of 15 meV) of approx. 7%, which does not substantially change with the pump power (Supplementary Note 3).

In order to provide a model to support our findings, we consider two spin-polarized excitonic reservoirs (in the GaAs QW and the monolayer), which are in turn generated by a circularly polarized optical excitation. This system can be described with a set of semi-classical Boltzmann equations, similar to those introduced in the Supplementary Note 4, but now also accounting for the spin-polarization of the excitonic reservoirs and condensed polaritons:

$$\frac{dN^{\pm}}{dt} = \left(W_1 n_1^{\pm} + W_2 n_2^{\pm}\right)(N^{\pm} + 1) - \frac{N^{\pm}}{\tau} - \frac{N^{\pm} - N^{\mp}}{\tau_s}, \quad (1)$$

$$\frac{dn_i^{\pm}}{dt} = P_i^{\pm} - W_{1(2)}^{\pm} n_i^{\pm}(N^{\pm} + 1) - \frac{n_i^{\pm}}{\tau_i} - \frac{n_i^{\pm} - n_i^{\mp}}{\tau_{s,i}} \quad (2)$$

Here $N^{\pm}$ and $n_i^{\pm}$ are the condensate and the reservoir densities in the two spin components, $W_i$ are the spontaneous scattering

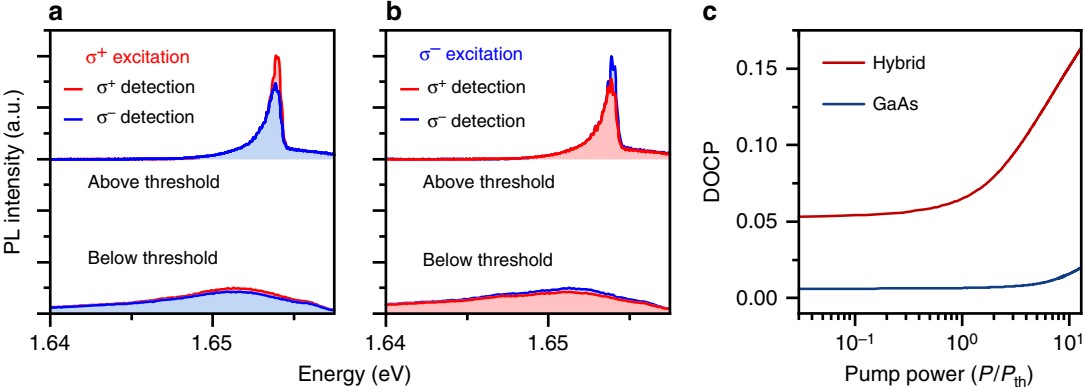

**Fig. 4** Polarized emission from the hybrid condensate. **a**, **b** Polarization resolved spectra at $k=0$ at a pump power of $10*P_{th}$ above and below the laser threshold for $\sigma^+$ (**a**) and $\sigma^-$ (**b**) excitation. The retained DOCP is 17.9% and 16.4%, respectively. In contrast, below the threshold only 9.8% ($\sigma^+$) and 7.2% ($\sigma^-$) are conserved. **c** Calculation of the pump power evolution of the degree of circular polarization of hybrid polaritons (red) and GaAs exciton (blue) polaritons

rates from the two reservoirs, $\tau$, $\tau_i$, $\tau_s$, and $\tau_{s,i}$ are the lifetimes and spin relaxation times of the condensate and the reservoirs, while $P_i$ are the pumping rates of the reservoirs, where $i=1$ corresponds to the QW and $i=2$ to the monolayer. We further assume that both reservoirs are created with a circularly polarized pumping:

$$P_i^- = 0; \quad P_i^+ = g_i P, \qquad (3)$$

where $g_1=1$ and $g_2=1(0)$ in the hybrid (QW) cavity case. We assume that the pumping is equally efficient in the monolayer and GaAs QW.

In Fig. 4c we plot the power evolution of the DOCP in the hybrid polariton system (red), in comparison with the pure GaAs system (blue). We note that the DOCP arises from a subtle interplay between spin-valley relaxation and bosonic condensation in the system. While, in the case of hybrid polaritons, the polariton pseudospin is better protected from depolarization by the effects of spin-valley locking, this effect is strongly enhanced by the bosonic amplification, which speeds up the relaxation dynamics from the reservoir, as reflected in our experiment.

## Discussion

In conclusion, we studied the density dependence of hybrid exciton-polaritons arising in a microcavity with four embedded GaAs QWs and a single monolayer of MoSe$_2$. The formation of a condensate of exciton-polaritons is clearly manifested by the strong nonlinearity in the input–output characteristics occurring at pump powers as low as 4.8 pJ per pulse, the collapse of the emission linewidth of the hybrid mode, and characteristic change of the blueshift above threshold signifying macroscopic occupation of the polariton mode. We demonstrate the effect of spin-valley locking in our condensate, a substantial feature inherent from the atomic monolayer, which paves the way to uniquely study valleytronic physics with bosonic condensates. We further believe that our work paves the way towards highly efficient, ultra-compact polariton-based light sources and valleytronic devices based on bosonic quantum fluids hosted in atomically thin materials, which ultimately can be operated at room temperature.

## Methods

**Experimental setup**. We used an optical setup similar to the one originally described in ref.[32] in which both spatially (near-field) and momentum-space (far-field) resolved spectroscopy and imaging are accessible. The sample is held at cryogenic temperatures in a helium flow cryostat. PL is collected through a 0.7 NA microscope objective lens and directed into an imaging spectrometer with 1200 groves mm$^{-1}$ grating via a set of relay lenses, projecting the proper projection plane onto the monochromator's entrance slit. The system's angular resolution is ~0.05 μm$^{-1}$ (~0.5°) and the spectral resolution is ~0.050 meV with a nitrogen-cooled Si-CCD serving as a detector. The error bars in Fig. 3e and the inset in Fig. 3f take into account the spectral resolution and the fit error. In Fig. 3f error bars are smaller than the data points and not shown.

**Data availability**. The data that support the findings of this study are available from the corresponding author upon reasonable request.

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

## Acknowledgements

We acknowledge fruitful discussions and support on the experiment by Mike Fraser. C.S. acknowledges support by the ERC (Project unLiMIt-2D), and the DFG within the Project SCHN1376 3-1. The Würzburg group acknowledges support by the State of Bavaria. A.N. and E.C. acknowledge the support from the megagrant 14.Y26.31.0015 and Goszadanie no. 3.2614.2017/4.6 of the Ministry of Education and Science of Russian Federation. A.V. K. acknowledges the support from the St-Petersburg State University in framework of the project 11.34.2.2012. S.H. and A.V.K. are grateful for funding received within the EPSRC Hybrid Polaritonics programme grant (EP/M025330/1). S.K. acknowledges the European Commission for the H2020 Marie Skłodowska-Curie Actions fellowship (Topopolis). S.T acknowledges support from NSF DMR 1838443 and NSF DMR 1552220.

## Author contributions

C.S and S.H. initiated the study and guided the work. M.Wu., N.L., and C.S. designed the Tamm device. N.L. and M.Wu. exfoliated, identified, and transferred the monolayer. M. Wu. fabricated the Tamm structure. H.C. grew the monolayer crystal. N.L., M.W., S.K., M.K., and E.E. performed experiments. N.L., M.K., M.W., S.K., S.B., C.S., and E.A.O. analyzed and interpreted the experimental data, supported by all coauthors. E.C, A.N., and A.V.K. provided the theory. C.S., M.W., N.L., S.K., and E.C. wrote the manuscript, with input from all coauthors.

## Additional information

**Competing interests:** The authors declare no competing interests.

