## [Peer Review File · Nature Communications]

Reviewers' comments:

Reviewer #1 (Remarks to the Author):

In the Article "Observation of bosonic condensation in a hybrid monolayer MoSe₂-GaAs microcavity", M. Waldherr et al. report on the nonlinear physics of a device combining TMD monolayer excitons and semiconductor quantum well excitons inside an optical microcavity. The entities formed out of this device are, in the strong coupling regime, "hybrid exciton-polaritons". The matter part of these hybrid polaritons thus inherits both from the TMD and quantum well excitons. The authors first report on the linear spectroscopy of their device and demonstrate the formation of a confined lower polariton mode. Increasing the optical power, they enter the nonlinear regime and observe polariton lasing in their structure. Finally, they discuss the degree of circular polarization of the emitted light which and its connection to valley polarization in the TMD. Important research efforts are currently being dedicated to the study of the nonlinear optical properties of TMD monolayers as well as TMD exciton polaritons. Examples of such studies include coherent nonlinear spectroscopy in MoSe₂ monolayers [1], room temperature lasing from MoTe₂ integrated with a silicon nanobeam cavity [2] or the observation of power-dependent energy-shifts in MoSe₂ under cw or pulsed excitation [3]. In the regime of strong coupling regime of cavity QED, TMD exciton polaritons are expected to show bosonic condensation. This effect has not yet been observed, and it represents an important challenge for the community. As a first step towards condensation of TMD exciton polaritons, I believe the present work is of interest for the community. However, before being able to recommend it for publication in Nature Communications, I would suggest some important clarifications to be added to the text regarding the physical mechanisms at work and the message of the paper.

Introduction

The introduction is well written. Two minor comments:

Typo in paragraph 3: "properties of exciton-polaritons critically depend".

Paragraph 4: The authors explain that the regime of bosonic condensation of TMD exciton polaritons has not been accessible yet due to "experimental" limitations. A few words (and maybe a reference?) to specify these limitations will be helpful to contextualize this work.

Paragraph 5: the use of "collective coupling" may lead to confusion, and "simultaneous coupling" for example may be more appropriate.

Hybrid polaritons

The demonstration of formation of confined hybrid polaritons in the system is convincing. However, some clarifications are needed.

Fig.1c shows the reflectivity spectrum prior to capping the structure. What are the fitted linewidths for quantum well excitons and TMD excitons? Both lines look relatively broad compared to state of the art samples, and more importantly, they seem to have very similar linewidths (of the order of 10 meV for both QW and TMD excitons). I find this problematic since it then contradicts the sentence "the hybridization leads to a longer radiative lifetime of exciton and reservoirs" depending on what hybridization the authors are referring to (between the excitons and a cavity photon, in which case I am fine? Or between GaAs excitons and TMD excitons, in which case there is a problem since both excitons have similar linewidths?). This needs clarification.

Why are the axis cut in energy at 1.66eV in Fig.2a and 1.665eV in Fig.2c? Can we see any luminescence from the higher energy polariton modes? Even in the case where the oscillator strength is too low, it would be better to show the full spectrum (all the way to 1.68eV) so the energy axes compare well with the ones used in Fig.2b and Fig.2d.

Polariton lasing

The experimental evidence for polariton lasing shown in Fig.3 (superluminal increase of the transmitted power, narrowing of the polariton line, and blueshift above threshold) is rather convincing. However what is missing here in my opinion is a good physical explanation of the mechanism at play.

It is known that an ensemble of 4 GaAs QWs embedded in a microcavity do not show polariton condensation. This is confirmed by the authors in the supplemental material as they show that increasing the optical power in the vicinity of the TMD flake leads to the transition to weak

coupling regime, followed by photon lasing. The authors also mention that polariton condensation of TMD exciton polaritons has not been observed so far for the range of experimental parameters used in this work. How does combining two systems that do not show polariton condensation when studied independently under the same experimental conditions lead to polariton condensation? In other words, starting from 4 GaAs QWs embedded in microcavity, how does coupling to a TMD flake allows observing condensation (knowing that demonstration of polariton condensation in GaAs QW based systems required the use of a minimum of 12 QWs)? The authors only give a vague explanation in paragraph 5 ("hybridization facilitates the build-up of the critical population for bosonic stimulation in the ground state"). They should clarify that point in order to fully support the claim of polariton condensation.

The number of points above threshold in Fig.S1 is very limited. In particular, the last three points in Fig.S1f show an unexpected, sizable blue shift. Can the authors confirm (and show?) that the energy stabilizes for even larger powers?

Typo in paragraph 10: "rapid drop in polariton linewidth from 2.1 meV".

Polarization properties

The authors finally measure the degree of polarization of the emitted light. In the linear regime, the authors measure a DOCP of 7% for GaAs exciton polaritons and 9% for the hybrid mode. In the linear regime, this suggests that the presence of the TMD flake only contributes in a minor way to the DOCP since GaAs alone already preserves a sizable degree of polarization (which can obviously not be associated with any valley index). As a consequence, I do not think the sentence "the polariton pseudospin is substantially better protected from depolarization by the effects of spin-valley locking" is an accurate conclusion for this measurement. Due to the optical selection rules in the TMD, the authors may of course associate the 9% DOCP to some remaining degree of valley polarization, but the authors should clarify the message and the conclusions in this part of the paper. Extending the coupled oscillator model will also allow to separate the different contributions to the DOCP in order to be more quantitative.

In the nonlinear regime, the authors observe an increased DOCP of the hybrid polariton light. By comparison, the authors show that, for the same powers, the DOCP of light emitted by pure GaAs exciton polaritons remains lower, of the order of 7%. I am not sure that this comparison is very relevant since the hybrid polaritons are well in the nonlinear regime (above threshold) whereas the GaAs QW exciton polaritons are still in the linear regime (below threshold). As a matter of fact, in the nonlinear regime, light emitted by pure GaAs polariton condensates would also show large degrees of polarization (see the onset of increasing DOCP for pure GaAs at the highest optical powers in the simulation Fig.4c). A more relevant comparison would thus require normalizing the power axes to the lasing thresholds...

This section thus needs rewording so as to clarify the message of the paper and to provide a more accurate description of the results.

References

1. T. Jakubczyk et al., Nano Lett. 16 (9) (2016)
2. Y. Li et al., Nature Nanotechnology 12 (2017)
3. G. Scuri et al., Phys. Rev. Lett. 120 (2018)

Reviewer #3 (Remarks to the Author):

Waldherr et al. present measurements of hybrid exciton-polariton condensation in MoSe₂. The work is timely and novel in that it uses applies methods of Tamm exciton-polaritons to observe polariton condensation in a hybrid monolayer microcavity. The phenomena have not been reported before in this or related materials, and it builds on the literature of the last several years with new observations. Since the work provides new observations that exploits an approach not currently in the literature, this work is appropriate for publication in a journal such as Nature Communications. Based on a previous review, the authors have adequately responded to some technical criticisms that could prevent publication. Some additional comments follow:

- Error bars have been added in some cases. It could be explained in the methods what the error bars correspond to. It looks like they might be fit error, but it is not clear if they take into account the randomness of the data collection itself. This might suggest the full experimental uncertainty is larger. Thus, it would be good to know what exactly is being plotted in caption or methods. Same applies to the Supplementary.

- I think the paper would be stronger with support of the effect from more than a single device. This is particularly true in the case of the apparently noisier data in Fig. 3e,f. If these trends were repeated, they would lend credence to the interpretation as a sample-independent phenomenon. This issue does not necessarily preclude publication at this stage as the data from the one sample are well-analyzed, but it would be a more compelling experiment with additional samples to support the results and show that trends are repeatable and therefore due to the phenomena suggested.

- The authors have made technical improvements to their manuscript. In particular, they have addressed the issue of over-emphasized speculation on room temperature operation in abstract and intro. This was a major conceptual flaw in their previous presentation that is now removed.

In the following, we will provide a point-by-point response to the referee's comments:

Reviewer #1 (Remarks to the Author):

In the Article "Observation of bosonic condensation in a hybrid monolayer MoSe₂-GaAs microcavity", M.Waldherr et al. report on the nonlinear physics of a device combining TMD monolayer excitons and semiconductor quantum well excitons inside an optical microcavity. The entities formed out of this device are, in the strong coupling regime, "hybrid exciton-polaritons". The matter part of these hybrid polaritons thus inherits both from the TMD and quantum well excitons. The authors first report on the linear spectroscopy of their device and demonstrate the formation of a confined lower polariton mode. Increasing the optical power, they enter the nonlinear regime and observe polariton lasing in their structure. Finally, they discuss the degree of circular polarization of the emitted light which and its connection to valley polarization in the TMD.

Important research efforts are currently being dedicated to the study of the nonlinear optical properties of TMD monolayers as well as TMD exciton polaritons. Examples of such studies include coherent nonlinear spectroscopy in MoSe₂ monolayers [1], room temperature lasing from MoTe₂ integrated with a silicon nanobeam cavity [2] or the observation of power-dependent energy-shifts in MoSe₂ under cw or pulsed excitation [3]. In the regime of strong coupling regime of cavity QED, TMD exciton polaritons are expected to show bosonic condensation. This effect has not yet been observed, and it represents an important challenge for the community. As a first step towards condensation of TMD exciton polaritons, I believe the present work is of interest for the community. However, before being able to recommend it for publication in Nature Communications, I would suggest some important clarifications to be added to the text regarding the physical mechanisms at work and the message of the paper.

Our response: We thank the referee for the positive assessment of our work and his appreciation of its importance. We address the remaining points of his/her report below.

Introduction

The introduction is well written. Two minor comments:

Typo in paragraph 3: "properties of exciton-polaritons critically depend".

Our response: We fixed the typo.

Paragraph 4: The authors explain that the regime of bosonic condensation of TMD exciton polaritons has not been accessible yet due to "experimental" limitations. A few words (and maybe a reference?) to specify these limitations will be helpful to contextualize this work.

Our response:

The reservoir dynamics in monolayer TMDCs take place on timescales of the order of few tens of picoseconds (see Lundt et al., 2D materials, 025096, 2017), which complicates the build-up of a sufficiently populated exciton-reservoir. This, in turn, complicates bosonic scattering into the ground state, required for polariton condensation.

Further, in the present generation of microcavities used in the field, the photon lifetime is lower than in traditional III-V DBR-based samples, which complicates to cross the quantum degeneracy threshold in the system. These factors are the current challenges in the field, which we are addressing with the approach of hybrid polaritons.

We have added an according statement in the manuscript in this context.

Paragraph 5: the use of “collective coupling” may lead to confusion, and “simultaneous coupling” for example may be more appropriate.

Our response: The terminology of collective strong coupling making clear that both active media together could strongly to the cavity mode is well established in the community, thus we would like to keep this wording.

Hybrid polaritons

The demonstration of formation of confined hybrid polaritons in the system is convincing. However, some clarifications are needed.

Fig.1c shows the reflectivity spectrum prior to capping the structure. What are the fitted linewidths for quantum well excitons and TMD excitons? Both lines look relatively broad compared to state of the art samples, and more importantly, they seem to have very similar linewidths (of the order of 10 meV for both QW and TMD excitons).

Our response:

The fitted FWHM for MoSe₂ and GaAs are 10.1 meV and 10.0 meV, respectively. The linewidth of MoSe₂ is comparable to literature values (e.g. 5 meV on SiO₂, see J. Wierzbowski et al, Scientific Reports 7, 12383 (2017)) without encapsulation (O. Iff et al, Optica 4, 6 pp669-673 (2017)). The quantum well exciton's linewidth of 10 meV is somewhat broader than state of the art QWs, however, we note that our QWs are rather narrow (5 nm thickness), and thus strongly affected by inhomogeneous broadening effects. We clearly note, that neither in the case of the QW excitons, nor the monolayer excitons, the inhomogeneously broadened linewidth can be directly associated with their emission dynamics.

I find this problematic since it then contradicts the sentence “the hybridization leads to a longer radiative lifetime of exciton and reservoirs” depending on what hybridization the authors are referring to (between the excitons and a cavity photon, in which case I am fine? Or between GaAs excitons and TMD excitons, in which case there is a problem since both excitons have similar linewidths?). This needs clarification.

Our response:

As stated above, in both cases (TMDC and QW), the linewidth are inhomogeneously broadened. MoSe₂ lifetime has been directly measured via streak camera measurements to be <3 ps (G. Wang et al, Appl. Phys. Lett. 106, 112101 (2015)), whereas the lifetime of GaAs excitons exceeds hundreds of ps.

Yet, clearly, the hybridization occurs between excitons and the cavity photon, whereas no direct coupling between the excitons is present in our system (see Hamiltonian in supplementary section 1).

Why are the axis cut in energy at 1.66eV in Fig.2a and 1.665eV in Fig.2c? Can we see any luminescence from the higher energy polariton modes? Even in the case where the oscillator strength is too low, it would be better to show the full spectrum (all the way to 1.68eV) so the energy axes compare well with the ones used in Fig.2b and Fig.2d.

Our response:

Data at higher energies is not available, because the laser excitation is at 741 nm (1.673 eV) with a width of ~ 1 nm due to 2 ps pulses. Further, the axis ranges differ, because the two separate measurements were taken with different grating positions in the spectrometer.

Polariton lasing

The experimental evidence for polariton lasing shown in Fig.3 (superluminal increase of the transmitted power, narrowing of the polariton line, and blueshift above threshold) is rather convincing. However what is missing here in my opinion is a good physical explanation of the mechanism at play. It is known that an ensemble of 4 GaAs QWs embedded in a microcavity do not show polariton condensation. This is confirmed by the authors in the supplemental material as they show that increasing the optical power in the vicinity of the TMD flake leads to the transition to weak coupling regime, followed by photon lasing.

Our response:

This is correct, indicating that in our case exciton- polariton relaxation in GaAs quantum wells is not efficient enough to allow for the polariton lasing in our Tamm-device. The relaxation is too slow compared to the radiative recombination rates, so that polariton condensates cannot be formed at the pumping intensities below the excitonic Mott transition. On the other hand, in TMD flakes the excitonic Mott transition is pushed to much higher densities compared to GaAs quantum wells due to the order of magnitude smaller exciton Bohr radius. Hybrid polaritons formed by TMD and GaAs excitons combine advantages of the high Mott density (TMD) and strong exciton-exciton scattering (GaAs). This combination is crucial for polariton lasing in our structure.

The authors also mention that polariton condensation of TMD exciton polaritons has not been observed so far for the range of experimental parameters used in this work. How does combining two systems that do not show polariton condensation when studied independently under the same experimental conditions lead to polariton condensation? In other words, starting from 4 GaAs QWs embedded in microcavity, how does coupling to a TMD flake allows observing condensation (knowing that demonstration of polariton condensation in GaAs QW based systems required the use of a minimum of 12 QWs)? The authors only give a vague explanation in paragraph 5 (“hybridization facilitates the build-up of the critical population for bosonic stimulation in the ground state”). They should clarify that point in order to fully support the claim of polariton condensation.

Our response:

The threshold of the hybrid polariton laser can be quantified by using the model introduced in SM (see eq. S1 and S2). Assuming that both reservoirs are pumped at the same rates ($g_1/g_2=1$ in our notation), the threshold reservoir injection rate for the hybrid condensation obeys the condition:

$$1/P = \tau[W_1\tau_1 + W_2\tau_2] = 1/P_{\{QW\}} + 1/P_{\{TMD\}}$$

This means that the pump threshold in the hybrid structure P is always lower than that in a QW or TMD layer alone. Physically this is due to the fact that both reservoirs scatter into the same polariton state, so the reservoir densities that is required to overcome polariton decay (this is the threshold condition) are much lower. We have added a short passage to the supplementary section of the manuscript.

We further would like to point out, that we do not want to rule out, that our system would promote polariton condensation in the monolayer without the presence of the GaAs QWs. Unfortunately, we cannot test this scenario in our device, as it is not possible to remove the QWs underneath the monolayer. Similar structures without GaAs QWs are currently built and investigated in our laboratory, but we believe that this is beyond the scope of this work.

The number of points above threshold in Fig.S1 is very limited. In particular, the last three points in Fig.S1f show an unexpected, sizable blue shift. Can the authors confirm (and show?) that the energy stabilizes for even larger powers?

Our response:

The input power at threshold in the reference measurement on bare GaAs QW-Tamm device in the supplementary material is very close to the damage threshold of our structure, and close to the peak power of our pump laser in our experimental setting. This makes investigation of higher powers not feasible. Yet, rather than observing a consistent blueshift, the last data points rather fluctuate close to the empty cavity energy. More importantly, the observed threshold is two orders of magnitude higher in contrast to the hybrid structure in the main text, which we believe leaves little doubts about attributing the observed effect to standard cavity lasing in the weak coupling regime.

Typo in paragraph 10: “rapid drop in polariton linewidth from 2.1 meV”.

Our response: We thank the referee for pointing out the typo and corrected it in the manuscript.

Polarization properties

The authors finally measure the degree of polarization of the emitted light. In the linear regime, the authors measure a DOCP of 7% for GaAs exciton polaritons and 9% for the hybrid mode. In the linear regime, this suggests that the presence of the TMD flake only contributes in a minor way to the DOCP since GaAs alone already preserves a sizable degree of polarization (which can obviously not be associated with any valley index). As a consequence, I do not think the sentence “the polariton pseudospin is substantially better protected from depolarization by the effects of spin-valley locking” is an accurate conclusion for this measurement. Due to the optical selection rules in the TMD, the authors may of course associate the 9% DOCP to some remaining degree of valley polarization, but the authors should clarify the message and the conclusions in this part of the paper. Extending the coupled oscillator model will also allow to separate the different contributions to the DOCP in order to be more quantitative.

Our response:

We note, that the statement “the polariton pseudospin is substantially better protected from depolarization by the effects of spin-valley locking” cited by the referee clearly was refereeing to model on the non-linear regime of our device, and we believe it is reasonable. However, we have toned down the statement in the main text, and made clear this conclusion reflects our modelling.

In fact, we cannot see how a reliable quantitative modelling of the spin polarization can be carried out within the coupled oscillator approach. The spin-dependent Boltzmann rate equations, which we apply in our analysis in our paper, is a much more conventional tool, and we are confident that our theoretical analysis is sufficiently thorough to disentangle the effects stemming from spin-polarized GaAs polaritons and spin-valley locked TMDC polaritons.

In the nonlinear regime, the authors observe an increased DOCP of the hybrid polariton light. By comparison, the authors show that, for the same powers, the DOCP of light emitted by pure GaAs exciton polaritons remains lower, of the order of 7%. I am not sure that this comparison is very relevant since the hybrid polaritons are well in the nonlinear regime (above threshold) whereas the GaAs QW exciton polaritons are still in the linear regime (below threshold). As a matter of fact, in the nonlinear regime, light emitted by pure GaAs polariton condensates would also show large degrees of polarization (see the onset of increasing DOCP for pure GaAs at the highest optical powers in the simulation Fig.4c). A more relevant comparison would thus require normalizing the power axes to the lasing thresholds... This section thus needs rewording so as to clarify the message of the paper and to provide a more accurate description of the results.

Our response:

We agree with the referee that a more prudent comparison would be a pure GaAs Tamm-plasmon polariton condensate, which we unfortunately do not observe in our sample, and, as far as we know, is not being reported on in the literature either.

We already state in the main text that the GaAs is in the linear regime. In order to account for the referee's comment, we chose a more careful wording in the main text.

References

1. T. Jakubczyk et al., Nano Lett. 16 (9) (2016)
2. Y. Li et al., Nature Nanotechnology 12 (2017)
3. G. Scuri et al., Phys. Rev. Lett. 120 (2018)

Reviewer #3 (Remarks to the Author):

Waldherr et al. present measurements of hybrid exciton-polariton condensation in MoSe₂. The work is timely and novel in that it uses applies methods of Tamm exciton-polaritons to observe polariton condensation in a hybrid monolayer microcavity. The phenomena have not been reported before in this or related materials, and it builds on the literature of the last several years with new observations. Since the work provides new observations that exploits an approach not currently in the literature, this work is appropriate for publication in a journal such as Nature Communications. Based on a previous review, the authors have adequately responded to some technical criticisms that could prevent publication. Some additional comments follow:

Our response:

We thank the referee for his/her recommendation to publish the paper and try to address his remaining comments in the following.

- Error bars have been added in some cases. It could be explained in the methods what the error bars correspond to. It looks like they might be fit error, but it is not clear if they take into account the randomness of the data collection itself. This might suggest the full experimental uncertainty is larger. Thus, it would be good to know what exactly is being plotted in caption or methods. Same applies to the Supplementary.

Our response:

We have followed the referees suggestions and added a corresponding statement in the methods section.

- I think the paper would be stronger with support of the effect from more than a single device. This is particularly true in the case of the apparently noisier data in Fig. 3e,f. If these trends were repeated, they would lend credence to the interpretation as a sample-independent phenomenon. This issue does not necessarily preclude publication at this stage as the data from the one sample are well-analyzed, but it would be a more compelling experiment with additional samples to support the results and show that trends are repeatable and therefore due to the phenomena suggested.

Our response:

We are glad that the referee acknowledges that the current data is sufficient to substantiate the claim of this paper. Further, we share the interest in a larger number of devices and we are indeed developing a new generation of hybrid Tamm-plasmon cavities. However, we are convinced that the referee can appreciate that development and manufacturing of such kind of samples is very demanding and time consuming.

The fabrication of more samples with improved techniques (including encapsulated monolayers) is currently done, however preliminary data (which indeed also indicated strongly superlinear emission from the hybrid mode) are too in-mature to be included in this publication.

- The authors have made technical improvements to their manuscript. In particular, they have addressed the issue of over-emphasized speculation on room temperature operation in abstract and intro. This was a major conceptual flaw in their previous presentation that is now removed.

Our response: We thank the referee for his appraisal of our continuous effort to improve our manuscript.

REVIEWERS' COMMENTS:

Reviewer #1 (Remarks to the Author):

Following the referees' comments, the authors have appropriately rephrased and improved parts of the paper. All the comments have been taken into account and I can now strongly recommend publication in Nature communications.

To rapidly come back to the comment on the sentence "the hybridization leads to a longer radiative lifetime", I would really recommend to write "the hybridization with cavity photons leads to longer radiative lifetime". I understand that this is implicit because the excitons are not directly coupled. However in the rest of the paper, the adjective "hybridd" refers to the mixed TMD/GaAs nature of the device, and this may lead to confusion.